

# The effects of exercise on FGF21 in adults: a systematic review and meta-analysis

Chuannan Liu[1,*], Xujie Yan[1,*], Yue Zong[1], Yanan He[1], Guan Yang[2], Yue Xiao[1] and Songtao Wang[1]

[1] School of Physical Education and Sports Science, South China Normal University, Guangzhou, China
[2] School of Physical Education, South China University of Technology, Guangzhou, China
[*] These authors contributed equally to this work.

## ABSTRACT

**Background.** Fibroblast growth factor 21 (FGF21) is a key hormone factor that regulates glucose and lipid homeostasis. Exercise may regulate its effects and affect disease states. Therefore, we sought to determine how exercise affects FGF21 concentrations in adults.

**Methods.** The review was registered in the International Prospective Systematic Review (PROSPERO, CRD42023471163). The Cochrane Library, PubMed, and Web of Science databases were searched for studies through July 2023. Studies that assessed the effects of exercise training on FGF21 concentration in adults were included. The random effect model, data with standardized mean difference (SMD), and 95% confidence intervals (CI) were used to evaluate the pooled effect size of exercise training on FGF21. The risk of heterogeneity and bias were evaluated. A total of 12 studies involving 401 participants were included.

**Results.** The total effect size was 0.3 (95% CI [$-0.3$–$0.89$], $p = 0.33$) when comparing participants who exercised to those who were sedentary. However, subgroup analysis indicated that concurrent exercise and a duration $\geq 10$ weeks significantly decreased FGF21 concentrations with an effect size of $-0.38$ (95% CI [$-0.74$–$-0.01$], $p < 0.05$) and $-0.38$ (95% CI [$-0.63$–$-0.13$], $p < 0.01$), respectively.

**Conclusion.** Concurrent exercise and longer duration may be more efficient way to decrease FGF21 concentrations in adults with metabolic disorder.

## INTRODUCTION

Fibroblast growth factor 21 (FGF21) is a recently discovered hormone that plays key roles in regulating energy homeostasis, glucose and lipid metabolism, and insulin sensitivity (*Geng et al., 2019*). It has potential applications in clinical medicine for obesity and related diseases such as cardiovascular diseases and type 2 diabetes (T2D) as it can independently predict aortic stiffness in patients with T2D (*Huang et al., 2021*). Drug development for these disease states has shifted focus from glucose metabolism to lipid metabolism, especially NAFLD, and analogs are now entering Phase 3 clinical research (*Talukdar & Kharitonenkov, 2021*).

Corresponding author
Songtao Wang,
wangsongtao@m.scnu.edu.cn

FGF21 is expressed in liver (*Nishimura et al., 2000*), muscle, pancreas, (*Fisher & Maratos-Flier, 2016*; *Oost et al., 2019*) and adipose tissues (*Zhang et al., 2008*) as well as in the brain where it crosses the blood–brain barrier (*Hung, Weihong & Abba, 2007*). FGF21 acts through FGF receptors (FGFRs) and its co-receptor, $\beta$-klotho (KLB), which is expressed (*Fisher & Maratos-Flier, 2016*) in the muscle, liver, pancreas, WAT (white adipose tissue), BAT (browning adipose tissue) and heart (*Fisher & Maratos-Flier, 2016*). FGF21 knock-out (KO) mice showed weight gain, body composition changes, and impaired glucose homeostasis (*Badman et al., 2009*) with the over-expression of FGF21 (*Fujii et al., 2019*). The exogenous administration of FGF21 in obese mice (*Hale et al., 2012*) may prevent or improve obesitywith an increased. Overweight, obesity, aging, T2D, and non-alcoholic fatty liver disease (NAFLD) are all associated with the progression of chronic metabolic diseases, especially those related to glucose-lipid metabolism. Paradoxically, FGF21 concentrations increase in conditions such as overweight, obesity, T2D (*Fève et al., 2016*) and NAFLD (*Headland, Clifton & Keogh, 2019*). These conditions are marked by high glucose and lipid levels (fat free acids, FFAs) with chronic inflammation, which may decrease FGFR or KLB expression in target tissues and lead to FGF21 resistance (*Fisher et al., 2010*; *Gallego-Escuredo et al., 2015*). Proteolytic cleavage may alter the form of FGF21 itself (*Markan, 2018*).

Regular physical activity may prevent numerous chronic diseases, including obesity, T2D and cardiovascular diseases (*Steven & Tim, 2004*) that are associated with metabolic disorders. Exercise has been shown to improve glucose homeostasis and lipid profiles, reduce fat mass (*Egan & Zierath, 2013*), increase arterial distensibility (*Currie, Thomas & Goodman, 2009*), and decrease inflammation in tissues and blood (*Metsios, Moe & Kitas, 2020*). However, the mechanisms underlying these changes are still not well understood. Studies have investigated the effect of exercise on circulating FGF21 levels and signaling pathways. Some studies have shown that moderate-to-vigorous intensity physical activity decreases serum FGF21 levels in older individuals (*Matsui et al., 2020*) and obese women (*Yang et al., 2011*). However, serum FGF21 increased in young health women after 2 weeks of exercise (*Cuevas-Ramos et al., 2012*). *Besse-Patin et al. (2014)* found that 8-week endurance training did not significantly change resting plasma concentrations of FGF21. Animal studies have determined that exercise increased FGF21 with 12 weeks of aerobic and resistance exercise (*Yang et al., 2019*). However, FGF21 decreased with 4 weeks of aerobic exercise (*Geng et al., 2019*). Thus, the effect of exercise on FGF21 in blood or tissues is still not well-understood.

A meta-analysis by *Khalafi et al. (2021)* found that acute exercise significantly increased circulating FGF21 levels in overweight and obese populations. Another review reported the effect of only aerobic exercise on the FGF21 protein or gene expression in overweight and obese individuals and animals (*Porflitt-Rodríguez et al., 2022*), emphasizing the mechanism of exercise and FGF21. In addition, most aerobic exercise studies reported that aerobic exercise decreased FGF21 levels (*Porflitt-Rodríguez et al., 2022*), while resistance training increased FGF21 levels (*Kruse et al., 2017*). The combination of aerobic exercise and resistance training decreased (*Motahari Rad et al., 2020*) or did not change FGF21 levels (*da Silveira Campos et al., 2018*). The effect of consistent exercise on FGF21 levels remains
unclear. Recently, *Kim et al. (2023)* determined that non-randomized controlled trials (RCTs) and RCT increased FGF21 levels, however their effect is not significant across different populations or types of exercise. This outcome was not in line with the current results and that analysis did not including both forms of exercise (*Kim et al., 2023*). Therefore, we conducted a systematic review and meta-analysis of the evidence on the effect of multiple types of exercise on FGF21 levels in adults to improve the clarity of the information in this field.

## METHODS

We followed the Preferred Reporting Items for Systematic Reviews and Meta-Analyses (PRISMA) statement. The protocol was registered in the International Prospective Systematic Review (PROSPERO, CRD42023471163).

### Search strategy

We searched in The Cochrane Library, PubMed, and Web of Science databases published up to July 2023 for all language articles using the search terms "FGF-21 OR fibroblast growth factor-21 OR fibroblast growth factor 21" AND "exercise OR training OR exercise training OR activity OR activities OR physical activity OR physical activities". Duplicate articles were removed using EndNote X9. Studies that were not related or non-experimental were removed based on their titles and abstracts. Two authors collected and synthesized data independently, and necessary information for our analysis. In case of any disagreement, a third author was consulted.

### Inclusion and exclusion criteria

The inclusion criteria were: (1) participants ≥ 18 years old; (2) only exercise intervention; (3) contorl group without exercise or other intervention; (4) detection of FGF21 in blood (serum or plasma) and (5) RCTs.

The exclusion criteria were: (1) proceedings article, non-full text, dissertations and reports; (2) animal studies; (3) non-RCTs; (4) uncontrolled and cross-sectional studies; (5) exercise and other pattern combined intervention; (6) acute exercise study and (7) no data.

### Data extraction

Data were extracted using a standardized assessment table that included the following categories: author, year, population (age, gender, BMI), sample, details of exercise and outcomes. Moreover, two studies (*Pérez-López et al., 2021*) depicted the data with the Get Data software in their graphics.

### Quality assessment

Two reviewers independently evaluated the quality of studies according to the inclusion standards. Review Manager 5.4 software was used, and any methodological quality discrepancies were discussed with the third reviewer. The quality assessment was executed according to the Cochrane criteria, following the item: random sequence generation, allocation concealment, blinding of participants and personnel, blinding of outcome

assessment, incomplete outcome data, selective reporting (for randomized controlled trials), and other biases.

## Statistical analyses

The results were continuous variables with mean and standard deviation (SD), and statistical analysis was conducted using Review Manager 5.4 with effect sizes (ES) were measured using mean, SD and sample size. Summary estimates with 95% confidence intervals (CI) were pooled applying the random effect model or fixed effect model. The degree of heterogeneity of the effect sizes was quantified using the I2 statistic as follows: <25% indicated low; 25–50% is moderate, 51–75% is substantial, and >75% is high heterogeneity.

Subgroup analyses followed exercise types, exercise intensity, duration, and exercise time, with the aim of discussing the effect of exercise on FGF21 in individuals in adults. The risk of publication bias was estimated using sensitivity analysis tests in Stata 15. A *p* value less than 0.05 was considered significant for all analyses. The detailed results of the statistical analyses can be found in the Supplementary Files.

# RESULTS

## Search results

We identified 1,899 articles in all databases, and after removing duplicates and screening titles and abstracts, 157 full-text articles were further reviewed for eligibility. A total of 12 randomized controlled trials were included in the meta-analysis, with a total of 18 eligible sets of data (Fig. 1).

## Characteristics of included studies

Table 1 shows the characteristics of the available articles. The studies included a total of 401 participants with obesity, overweight, aging, T2D, NAFLD, nonalcoholic steatohepatitis (NASH), metabolic syndrome, and post/premenopausal obese women with metabolic disorders. The control group was sedentary or had only light physical activity. The individuals range from 19 to 78 years old, and the mean body mass index (BMI) was greater than 24 kg/m$^2$; the highest being 60 kg/m$^2$. The FGF21 levels ranged from 128 to 600 pg/mL.

The studies included in the meta-analysis were RCTs that involved aerobic exercise (*Kong et al., 2016*; *Taniguchi et al., 2016*; *Banitalebi et al., 2019*; *Keihanian, Arazi & Kargarfard, 2019*; *Pérez-López et al., 2021*; *Haghighi et al., 2022*; *Stine et al., 2023*), resistance exercise (*Keihanian, Arazi & Kargarfard, 2019*; *Saeidi et al., 2019*; *Takahashi et al., 2020*; *Shabkhiz et al., 2021*; *Haghighi et al., 2022*), or combined exercises (aerobic and resistance) (*Banitalebi et al., 2019*; *Motahari Rad et al., 2020*; *Chang & Namkung, 2021*; *Pérez-López et al., 2021*). The intensity of aerobic exercise ranged from 45 to 95% of maximum heart rate (HRmax) or VO2 peak, the intensity of the resistance exercise was more than 55% of one-repetition maximum (1RM) for multiple movements (including squat, chest press, leg press, standing military press, knee extension, seated cable rowing, knee curl, biceps curl, standing calf raise, triceps press, back-extension, and abdominal crunch). A total of seven articles focused

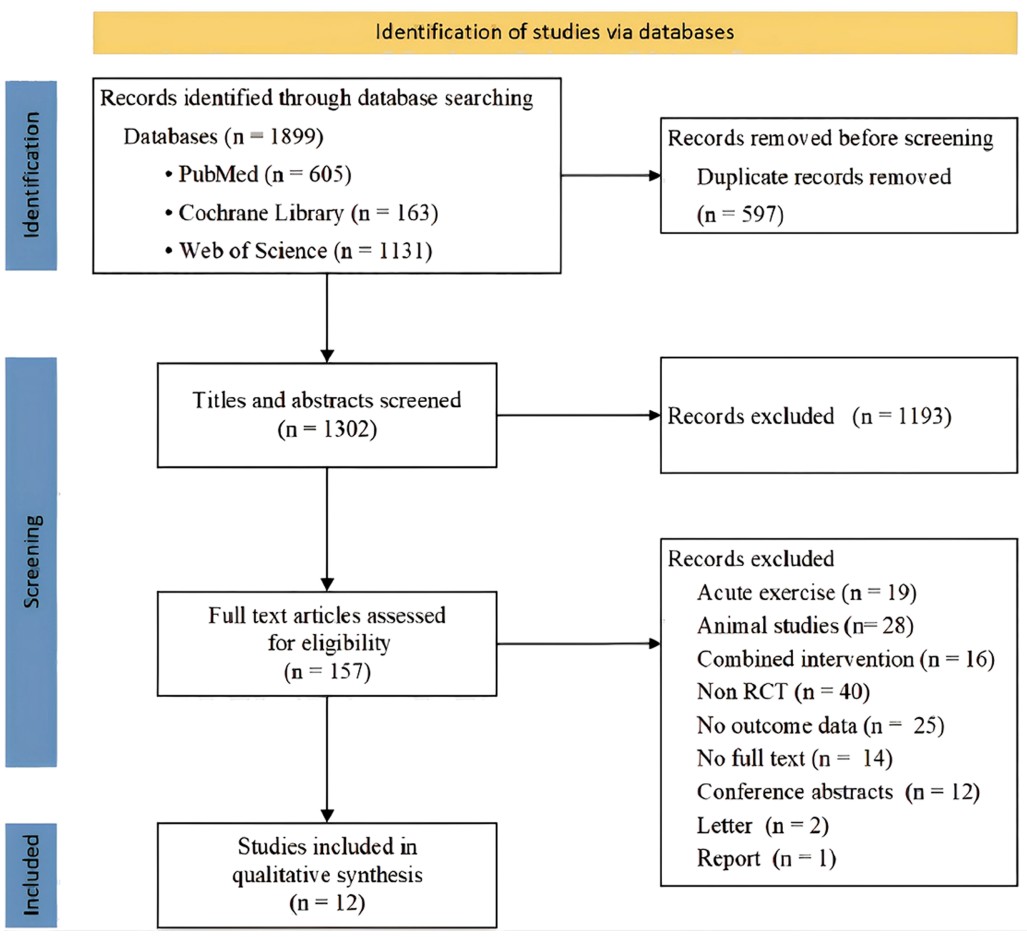

**Figure 1** Flow diagram of studies search, selection and inclusion process.

on an exercise duration of more than 10 weeks, and five studies included exercise periods of less than 10 weeks; training frequency was three or four times per week, with 30 to 60 min per session. Four available studies included an exercise intensity over 80%, 12 less than 80%, and two articles did not mention the exercise intensity.

Secondary outcomes included glucose (glucose, insulin, HbA1c, and Homeostatic Model Assessment of Insulin Resistance (HOMA-IR)), lipid (cholesterol, triglycerides, HDL, LDL, and FFAs), and anthropometric parameters (weight, BMI, and body fat percent). These data may be related to metabolic diseases and FGF21 (*Headland, Clifton & Keogh, 2019*).

## Risk of bias

Figure 2 shows the percentage risk of bias from all of the included studies. Most studies shown low and unclear risk in major standard. Low risk appeared in a high percentage of incomplete outcome, selective reporting and other bias. Studies (*Kong et al., 2016*; *Keihanian, Arazi & Kargarfard, 2019*; *Takahashi et al., 2020*; *Haghighi et al., 2022*) used single blinding had a high risk. Mostly, all the studies (*Kong et al., 2016*) were considered

Peer J

**Table 1  Characteristics of the studies and exercise intervention included in this meta-analysis.**

| Study | Population | N. | Age | BMI (kg/m²) | Intervention | | | | FGF21 (pg/mL) | Second outcomes | Exercise performance |
|---|---|---|---|---|---|---|---|---|---|---|---|
| | | | | | Exercise type | Duration | Frequency | Intensity | | | |
| Banitalebi et al. (2019) | T2D | 52 C:17 T1:17 T2:18 | 49.7–62.1 | 24.3–33.64 | T1: Con-E (AT (treadmill or cycle ergometer) +RT (8 motion)) T2: SIT exercise | 10 W | AT: 50 min/b RT: 12-10R for 2–3 sets SIT: 4b* 30s(), 3ds/w | AT: 70% HRmax RT: NR SIT: All-out | Serum C: 235.96 ± 69.74 T1:229.63 ± 94.89 T2:204.67 ± 111.36 | BF% n.s. Weight↓, BMI↓, Insulin↓, HbA1c↓, HOMA-IR↓, FPG↓†† | VO2 peak↑ for T1 |
| Chang & Namkung (2021) | Metabolic syndrome women | 42 C:30 T:12 | 45.3–68.1 | 25.9–33.9 | Combined endurance and strength exercise | 12 W | 50 min/b; 3dsw | NR | Serum C:219.7 ± 123.9 T:176.3 ± 107.4 | BMI↓, WC↓,B F%↓† | NR |
| Keihanian, Arazi & Kargarfard (2019) | T2DM men | 34 C:12 T1:11 T2:11 | 50.6–54.1 | 29.1–35.7 | AT (running); RT (7 motion) | 8 W | AT: 30-45 min/b RT: 45 min/b, 3ds/w; | AT: 75%–85% HRmax RT: 10RM-max | Serum C:128.2 ± 1.9 T1:250.1 ± 6.7 T2:184 ± 4.6 | FBS↓, Insulin↓, HbA1c↓, Cholesterol↓, Triglyceride↓, HDL↑, LDL↓, HOMA-IR↓†† | VO2 peak for T1/T2↑; Strength performance for T1/T2↑ |
| Kong et al. (2016) | Obese women | 18 C:10 T:8 | 17.8–22 | 23.8–28.6 | AT (cycling) | 5 W | HIIT: 20 min/b MICT: 40 min/b; 4ds/w | HIIT: 95% MICT: 65% VO2peak continuing training | Serum C:0.6 ± 0.6 T:0.5 ± 0.4 | Weight, BMI n.s. | VO2 peak↑ |
| Motahari Rad et al. (2020) | T2DM men | 51 C:17 T1:17 T2:17 | 41.7–47.7 | 27.8–31.3 | AT+RT RT+AT RT: 6 motion AT: (walking/running by treadmill) | 12 W | RT: 8-10R for 3 sets AT:10 min; 3ds/w | RT: 70–80% 1RMmax AT: 90–95% HRmax | Serum C:510.9 ± 106.1 T1:441.7 ± 110 T2:449 ± 98.8 | Body mass↓, BMI↓, HOMA-IR↓, HbA1c↓†† | VO2 peak for T1/T2↑; UB strength/ LB strength for T1/T2↑ |
| Pérez-López et al. (2021) | Post/pre menopausal women with obesity | 35 C:13 T1:10 T2:12 | 40.3–61.6 | 28.7–39.8 | AT (running on a treadmill) RT(6 motion) Con-E(AT+RT) | 12 W | AT: 60 min/b RT:8-12R for 3 sets Con-E:20 min AT + 40 min RT, 3ds/w | AT: 55–75%HRR RT: 65% 1RMmax Con-E: same as AT and RT | Serum C:205.2 ± 64.35 T1:146.09 ± 56.96 T2:196.52 ± 80 | Body mass n.s.; Glucose pro-file:(Glucose, insulin,HOMA-IR,HbA1c) n.s. Lipid pro-file:(Cholesterol, Triglyc-erides, HDL, LDL) n.s. BMI↓†† Body composition↓† for AE | NR |
| Saeidi et al. (2019) | Postmenopausal women | 24 C:12 T:12 | 51–63 | 25.4–30.1 | RT (12 motion) | 8 W | 30s for each session, 3ds/w | 55% 1RM-max. | Plasma C:252.7 ± 5.5 T:281.6 ± 5.5 | Body mass, BMI n.s. | NR |
| Shabkhiz et al. (2021) | Elderly men with/ without T2D | 44 C1:12 C2:10 T1:12 T2:10 | Without T2D: 66.78–77.38 With T2D: 71.85–73.05 | Without T2D: 23.45–31.49 With T2D: 22.62–29.68 | RT (7 motion) | 12 W | 10R/b for 3 sets, 3ds/w | 70% 1RM-max | Serum C1:285.05 ± 100.19 T1:253.24 ± 116.1 C2:403.22 ± 175.41 T2:324.07 ± 107.7 | Weight, BMI, In-sulin n.s. Glucose↓, HOMA-IR↓†† | Leg press strength for both T1/T2↑ |
| Takahashi et al. (2020) | Patients with NAFLD | 50 C:27 T:23 | 35.9–67.7 | 23.2–33.7 | RT (push-ups and squats) | 12 W | 3 sets for 20–30 min/b; 3ds/w | NR | Serum C:184.6 ± 113.3 T:142.9 ± 105.9 | BMI n.s. | NR |

*(continued on next page)*

**Table 1** (*continued*)

| Study | Population | N. | Age | BMI (kg/m²) | Intervention | | | | FGF21 (pg/mL) | Second outcomes | Exercise performance |
|---|---|---|---|---|---|---|---|---|---|---|---|
| | | | | | Exercise type | Duration | Frequency | Intensity | | | |
| *Taniguchi et al. (2016)* | Elderly men | 33 C:15 T:17 | 65.4–73.8 | 20.5–25.7 | AT (cycling) | 5 W | 30 min for 1–2 week, 45 min for 3–5 min, 3ds/w | 60%VO2max for first week, 70%VO2max for 2–3 week, 75%VO2max for 4-5 week. | Serum C:255.9 ± 88.5 T:218.5 ± 65.1 | BMI,BF%, FFA, Fasting insulin, HOMA-IR n.s. HbA1c↓[†] | VO2 peak↑ |
| *Haghighi et al. (2022)* | overweight and obesity | 30 C:10 T1:10 T2:10 | 30-45 | 29.41 ± 3.02 | AT(running) RT(8 motion) | 8W | AT: 15 min per session, RT:50-60 min per session; both 3ds/w | AT: 85–95% HRmax RT:85–95% 1RM | Serum C: 300.1 ± 24.2 T1:371.1 ± 66.49 T2:380.9 ± 49.08 | BMI n.s. Weight, FPB%↓[††] | VO2 peak – |
| *Stine et al. (2023)* | NASH | 24 C:8 T:16 | 25-69 | 32.8 ± 5.2 | AT | 20W | 30 min per session, 5ds/w | 45–55% VO2peak | Serum C:320.06 ± 354.3 T:240.06 ± 351.7 | BMI, WC, body fat n.s. Weight↓, HC↓[†] | VO2 peak – |

**Notes.**

AT, aerobic exercise; b, bout; BMI, body mass index; BF%, body fat percent; C, control group; Con-E, concurrent of aerobic and resistance exercise; d, day; F, female; FPG, fasting blood glucose; FBS, fasting blood sugar; DL, high-density lipoprotein; HIIT, high intensity interval training; HR, heart rate; HRR, heart rate reserve; LDL, low-density lipoprotein; MICT, moderate intensity continuous training; NAFLD, nonalcoholic fatty liver disease; NASH, Nonalcoholic Steatohepatitis; NR, not report; n.s., no significant; RM, repetition maximum; RT, resistance exercise; SIT, sprint interval training; T2D, type 2 diabetes; T, training group; VO2 peak, maximal oxygen consumption; W, week; WC, waist circumference.

[†] T1 is significant.

[††] T1/T2 are both significant ($p < 0.05$).
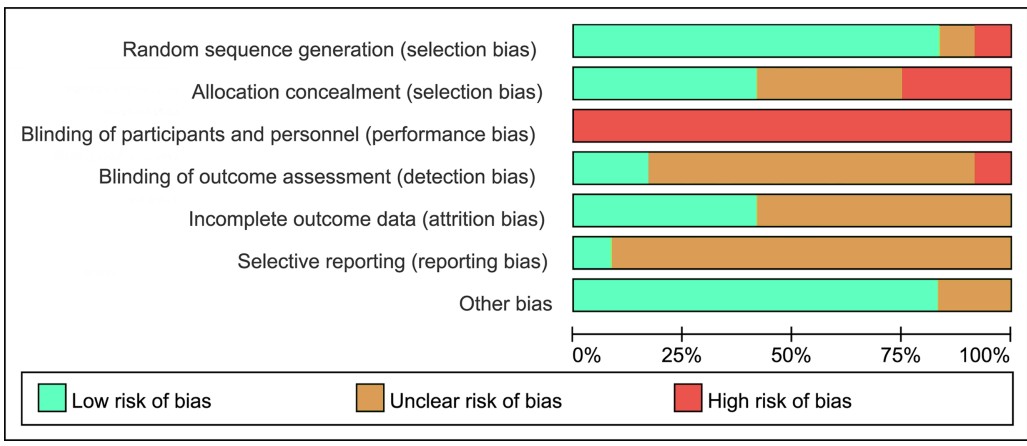

**Figure 2 Risk of bias of included studies.**

| Study or Subgroup | Experimental Mean | SD | Total | Control Mean | SD | Total | Weight | Std. Mean Difference IV, Random, 95% CI |
|---|---|---|---|---|---|---|---|---|
| Banitalebi | 217.15 | 102.28 | 28 | 235.96 | 69.74 | 14 | 8.3% | -0.20 [-0.84, 0.44] |
| Chang | 176.3 | 107.4 | 14 | 219.7 | 123.9 | 15 | 8.1% | -0.36 [-1.10, 0.37] |
| Haghighi | 376.3 | 57.07 | 20 | 300.1 | 24.2 | 10 | 7.7% | 1.51 [0.65, 2.38] |
| Keihanian | 215.61 | 34.22 | 23 | 128.2 | 1.9 | 11 | 7.2% | 3.01 [1.96, 4.05] |
| Kong | 0.5 | 0.4 | 10 | 0.6 | 0.6 | 8 | 7.5% | -0.19 [-1.12, 0.74] |
| Motahari Rad | 445.35 | 102.8 | 30 | 510.9 | 106.1 | 13 | 8.3% | -0.62 [-1.29, 0.05] |
| Perez-Lopez | 174.59 | 73.97 | 23 | 205.2 | 64.35 | 12 | 8.1% | -0.42 [-1.13, 0.28] |
| Saeidi | 281.6 | 5.5 | 12 | 252.7 | 5.5 | 12 | 5.1% | 5.07 [3.31, 6.84] |
| Shabkhiz | 253.24 | 116.1 | 10 | 285.05 | 100.19 | 12 | 7.8% | -0.28 [-1.13, 0.56] |
| Shabkhiz | 324.07 | 107.7 | 10 | 403.22 | 175.4 | 12 | 7.7% | -0.51 [-1.37, 0.34] |
| Stine | 240.06 | 354.3 | 12 | 320.6 | 354.3 | 8 | 7.6% | -0.22 [-1.12, 0.68] |
| Takahashi | 142.9 | 105.9 | 23 | 184.6 | 113.3 | 27 | 8.5% | -0.37 [-0.93, 0.19] |
| Taniguchi | 218.5 | 65.1 | 17 | 255.9 | 88.5 | 15 | 8.1% | -0.47 [-1.18, 0.23] |
| | | | | | | | | |
| **Total (95% CI)** | | | 232 | | | 169 | 100.0% | 0.30 [-0.30, 0.89] |

Heterogeneity: Tau² = 1.00; Chi² = 86.95, df = 12 (P < 0.00001); I² = 86%
Test for overall effect: Z = 0.98 (P = 0.33)

**Figure 3 Overall analysis forest plot of exercise on FGF21.**

to have a high risk of performance bias based on the lack of non-blinding of participants and personnel.

## Impact of exercise on FGF21 concentration

Overall, the meta-analysis found no significant effect of total exercise training on circulating levels of FGF21 after exercise training with high heterogeneity (SMD = 0.3, 95% CI [−0.3–0.89], $p = 0.33$) (Fig. 3).

We conducted subgroup analyses according to the exercise protocol factors (exercise type, duration, and intensity have available data) and analyzed the relevant data. Figure 4 shows the subgroup analysis of resistance exercise with a high heterogeneity (1.1.2: SMD = 2.01, 95% CI [0.23–3.8], $p = 0.03$). The concurrent practice of resistance and aerobic showed significant effects with low heterogeneity at the FGF21 level exercises (1.1.1: SMD =

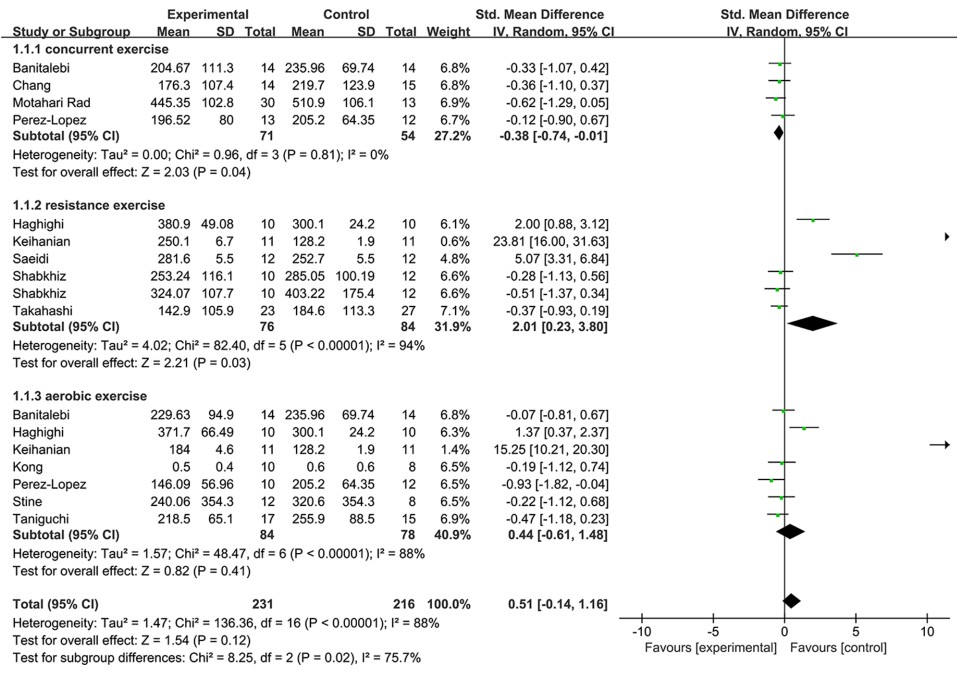

**Figure 4  Subgroup analysis of exercise type forest plot on FGF21.**

−0.38, 95% CI [−0.74 to −0.01], $p < 0.05$). In addition, aerobic exercise has no significant effect (1.1.3: SMD = −0.44, 95% CI [−0.61–1.48], $p = 0.41$).

Figure 5 shows the subgroup of exercise duration. Exercise durations ≥ 10 weeks were shown to decrease the FGF21 levels with low heterogeneity (1.2.1: SMD = −0.38, 95% CI [−0.63 to −0.13], $p < 0.01$). Exercise durations <10 weeks increased the FGF21 with high heterogeneity levels (1.2.2: SMD = 1.67, 95% CI [0.01–3.34], $p = 0.05$). Figure 6 shows the subgroup of exercise intensity and there were no significant effects from exercise intensities over 80% (1.3.1: SMD = 0.27, 95% CI [−0.91–1.45], $p = 0.43$) or less than 80% (1.3.2: SMD = 0.25, 95% CI [−0.42–0.92], $p = 0.47$).

## DISCUSSION

The review included 12 studies; we focused on the effects of exercise on FGF21 levels in adults as the results of previous studies have been inconclusive. Most available studies present improved lipid and glucose profiles and a significant increase VO2 peak. Aerobic exercise, resistance exercise, and a combination of both resistance and aerobic exercises were studied.

In this meta-analysis, exercise training had no significant effect on FGF21 with high heterogeneity across all available studies. Subgroup analysis found that exercise type of concurrent exercise (aerobic plus resistance) and the term of duration ≥ 10 weeks could significantly decrease FGF21 levels ($p < 0.05$) in adults, including those with with obesity, T2D, NASH, *etc*. Based on these findings, we unexpectedly observed an effect from exercise durations of over 10 weeks for the concurrent exercise period. Therefore,

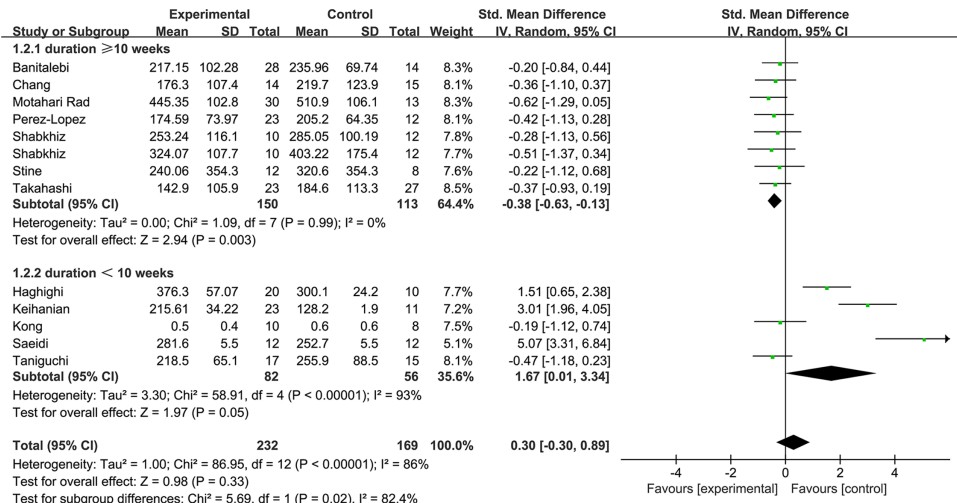

**Figure 5** Subgroup analysis of exercise duration forest plot on FGF21.

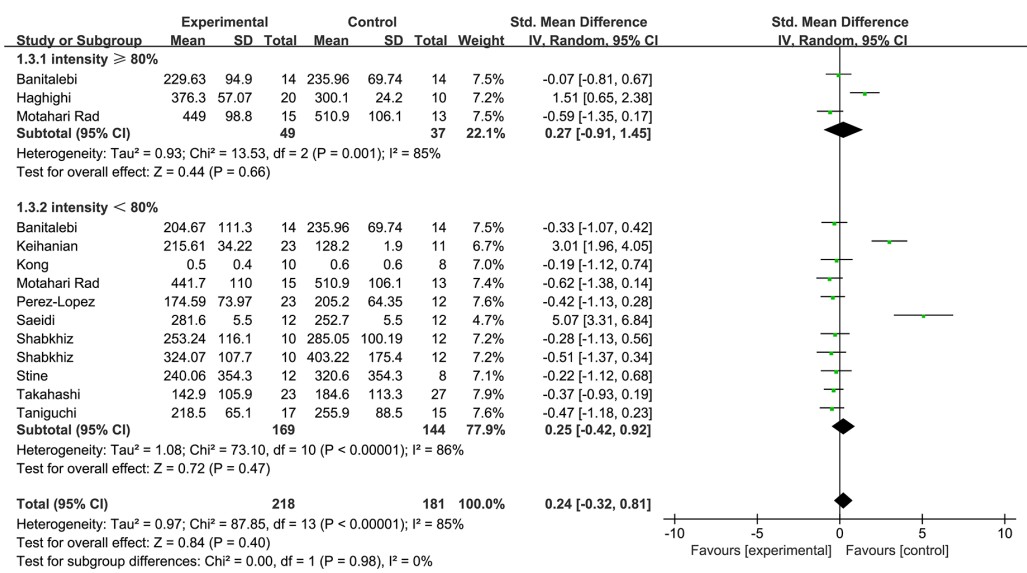

**Figure 6** Subgroup analysis of exercise intensity forest plot on FGF21.

using the proper exercise model or practicing a longer training duration may effectively decrease FGF21 concentrations, increase cardiovascular health and muscle strength, and decrease weight, BMI and body fat percent. This may, in turn, affect the glucose (glucose, insulin, HOMA-IR and Hb1Ac) and lipid profiles (cholesterol, triglycerides, HDL, LDL and FFAs). FGF21 levels are rarely low in healthy individuals, however, adults who are overweight or who have metabolic diseases have high levels. Therefore, a lower FGF21 level may indicate a more healthful state. Interestingly, resistance training and an exercise duration <10 weeks ($p < 0.05$) significantly increased FGF21 levels with high heterogeneity,

which was consistent with 2 week exercise interventions in healthy women (*Cuevas-Ramos et al., 2012*). This may be explained by the presence of a middle state between health and disease, like a sub-clinical stage or FGF21 compensatory response. The overall effects of single aerobic, resistance or different intensity outcomes of available studies showed no significant changes in FGF21 levels. Previous studies have reported its effects on promoting fatty acid oxidation, increasing energy expenditure, and improving insulin sensitivity (*Owen, Mangelsdorf & Kliewer, 2015*; *Angelin, Larsson & Rudling, 2012*), which are particularly important effects for individuals with metabolic disorders (*Kong et al., 2016*; *Taniguchi et al., 2016*; *Banitalebi et al., 2019*; *Keihanian, Arazi & Kargarfard, 2019*; *Saeidi et al., 2019*; *Matsui et al., 2020*; *Motahari Rad et al., 2020*; *Chang & Namkung, 2021*; *Shabkhiz et al., 2021*). However, these individuals have higher circulating levels of FGF21 than healthy individuals, which contradicts FGF21's physiological function (*Markan, 2018*). Furthermore, FGF21 levels have been linked to BMI, fat mass, waist circumference, and visceral adipose tissue, which may contribute to the elevation of circulating FFAs, glucose, and lipids (LDL and triglyceride) levels.

We know that aerobic exercise primarily benefits cardiovascular health and weight loss, while resistance training helps maintain muscle mass and strength, and may prevent osteoporosis and falls in menopausal women and the elderly (*Winters-Stone et al., 2013*). Moreover, American College of Sports Medicine (ACSM) recommends combining aerobic and resistance exercises, which may yield greater exercise benefits. Physical activity and structured exercise training are the cornerstone of therapy for various diseases (*Pedersen & Saltin, 2015*). Evidence from previous studies suggests that active individuals have fewer symptoms of T2D, including lower fasting blood sugar (FBS), lower LDL cholesterol, and homeostasis of glucose, lipids, triglycerides, and cholesterol (*Alberga et al., 2013*). Currently, endurance training is the primary mode of improving cardiovascular fitness by increasing VO2max (*Guadalupe-Grau et al., 2018*); resistance training aims to enhance neuromuscular connections and muscle strength (*Laurens, Bergouignan & Moro, 2020*). Both types of exercise have their advantages and disadvantages. However, researchers have recommended that a concurrent exercise program can provide greater improvement and prevent more adverse effects than a single-type program among individuals with obesity or the elderly (*Ferrari et al., 2016*).

Our systematic review and meta-analysis aimed to investigate the impact of exercise on FGF21 concentration in adults with chronic metabolic disorders. However, regarding the complex characteristics of population, age, gender, exercise protocols, or FGF21 timing, considerable heterogeneity ($I2 > 50\%$) was detected in most of the included studies. To address this issue, we performed a sub-analysis considering exercise type (aerobic, resistance, and concurrent exercise), exercise intensity (high intensity and middle or light intensity), and duration ($\geq 10$ weeks and $<10$ weeks). Although the population characteristics such as age and BMI had no set limit, we confined our subgroup analysis to intensity, exercise duration, and exercise type since the timing was approximately the same. We found that the subgroup results of intensity analysis did not significantly contribute to the heterogeneity.

Analysis showed that combined exercise training led to a marked decrease, consistent with the study by *da Silveira Campos et al. (2018)* that investigated moderate intensity combined exercise in obese women. However, single aerobic and resistance exercise training did not lead to significant changes in circulating FGF21, and we could not perform further meta-analysis due to insufficient data. Surprisingly, *Kruse et al. (2017)* found that resistance training for 10 weeks did not lead to significant changes in FGF21 concentration. Moreover, our analysis showed that the FGF21 concentrations did not change in young or older populations who engaged in habitual activity, nor did it change other hormones, such as adiponectin (*Lee et al., 2020*). Several studies included in our analysis reported conflicting results, with some studies reporting a significant decrease in FGF21 concentration (*Cuevas-Ramos et al., 2012*; *Willis et al., 2019*; *Yang et al., 2019*), while others reported a significant increase (*Fletcher et al., 2012*; *Geng et al., 2019*). We noted that the timing of blood collection was not always clear in these studies. It is worth mentioning that FGF21 has a circadian rhythm, and special stress (fasting, feed) can increase FGF21 concentrations (*Inagaki et al., 2007*; *Andersen, Beck-Nielsen & Højlund, 2011*). Therefore, this may contribute to the discrepancy in the reported outcomes.

The mechanisms underlying the decrease in FGF21 levels following aerobic, resistance, or combined exercises are dependent on the loss of fat mass in humans or the depletion of liver and adipose tissue, and the elevation of circulating FFAs that enhance fatty acid oxidation (*Bajer et al., 2015*), activated the peroxisome proliferator-activated receptor alpha (PPAR$\alpha$) pathway (*Yu et al., 2011*). One of the reasons for increased FGF21 levels following exercise is the production of more FGF21 by the muscles, along with other myokines such as growth differentiation factor 15, interleukin-6, and irisin (*Kim & Song, 2017*; *Motahari Rad et al., 2020*; *Chang & Namkung, 2021*), particularly at higher intensities with high FGF21. In addition, exercise can increase FGFRs and KLB expression in animals (*Geng et al., 2019*) , by decreasing FFAs, glucose, and TNF-$\alpha$ in target tissues enhancing the physiological effects FGF21 (*Salminen, Kaarniranta & Kauppinen, 2017*; *Salminen, Kauppinen & Kaarniranta, 2017b*). Conversely, individuals with obesity or type 2 diabetes have higher FGF21 concentrations and decreased expression of the FGFR1c receptor and co-receptor KLB/$\beta$-klotho (*Fisher et al., 2010*). Therefore, there should be greater focus on the expression of these receptors in future research (*Yang et al., 2019*). Notably, there are acute effects of exercise on FGF21 levels, however, under conditions of low receptor expression, this may exacerbate FGF21 resistance (*Shabkhiz et al., 2021*).

Despite the critical role of FGF21 in assessing health outcomes, this review and meta-analysis had some limitations. The available studies are relatively few, limiting the scope of further data meta-analysis. Our meta-analysis focused on FGF21 as a crucial factor in determining health outcomes. We listed the second outcomes but did not analyze their relationship and the available studies did not request precise exercise and blood sampling times to account for differences in outcomes. Publication bias may have been a factor since we excluded studies with grey literature, through our sensitivity analyses result was robust. Lastly, exercise training is a complex intervention and nutrition was not considered here, although nutrition affects FGF21 levels and health states, therefore, there are discrepancies in the baseline measurements of FGF21 levels.

## CONCLUSION

Our findings suggest that exercise training, particularly concurrent aerobic and resistance training over a long period (at least 10 weeks), may be a viable approach for improving FGF21 levels and health parameter outcomes in individuals with obesity, T2D, and NAFLD. Further research is needed to determine the timing, nutrition, and training patterns of FGF21 levels and to investigate the mechanisms underlying these effects in order to optimize programs for affected populations.

### Funding

The authors received no funding for this work.

### Competing Interests

The authors declare there are no competing interests.

### Author Contributions

- Chuannan Liu performed the experiments, analyzed the data, prepared figures and/or tables, authored or reviewed drafts of the article, and approved the final draft.
- Xujie Yan conceived and designed the experiments, performed the experiments, analyzed the data, prepared figures and/or tables, authored or reviewed drafts of the article, and approved the final draft.
- Yue Zong conceived and designed the experiments, prepared figures and/or tables, and approved the final draft.
- Yanan He conceived and designed the experiments, authored or reviewed drafts of the article, and approved the final draft.
- Guan Yang analyzed the data, authored or reviewed drafts of the article, and approved the final draft.
- Yue Xiao conceived and designed the experiments, authored or reviewed drafts of the article, and approved the final draft.
- Songtao Wang analyzed the data, authored or reviewed drafts of the article, and approved the final draft.

### Data Availability

The raw measurements are available in the Supplementary Files Raw data. The raw data shows the FGF21 level of experimental and control groups in those studies, which use the meta analysis.

### Supplemental Information

Supplemental information for this article can be found online at http://dx.doi.org/10.7717/peerj.17615#supplemental-information.

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
