# Peer review of "The effects of exercise on FGF21 in adults: a systematic review and meta-analysis"

_PeerJ, doi:10.7717/peerj.17615_

## Round 0.1 · original submission · Major Revisions

The manuscript is related to exercise and FGF21.

Some topics in the article should be clearly defined and addressed.

Revision is therefore required

Reviewer 1 ·

Basic reporting

The paper titled "The effects of exercise on FGF21 in adults: a systematic review and meta-analysis" by Chuannan Liu and colleagues presents a comprehensive review and meta-analysis examining the impact of exercise on Fibroblast Growth Factor 21 (FGF21) levels in adults. The study aims to clarify the relationship between various exercise regimens and FGF21 concentrations, a hormone associated with glucose and lipid metabolism. The authors conducted a detailed analysis including randomized controlled trials (RCTs) that reported on the effects of exercise on FGF21 levels. Their findings suggest that specific types of exercise, particularly concurrent exercise and exercises exceeding ten weeks in duration, significantly decrease FGF21 levels in adults with metabolic disorders, indicating a potential for healthier metabolic states.

Experimental design

The data collection and presentation of data seems reasonable.

Validity of the findings

Findings seem to be reasonable.

Additional comments

For this review, languages should be examined more carefully. For example, in the beginning of the abstract "we aim to analysis how exercise effect FGF21" should be "aim to analyze". Similar grammatical mistakes need to be checked and avoided for the manuscript.
Also in the initial page, some of the formatting are not correct: "The total effect size was 0.44 (95%CI [-0.18 ÿ 1.06], p = 0.16) compared exercise to sedentary." The dash has been misformatted. Please correct accordingly.

Reviewer 2 ·

Basic reporting

The meta-analysis entitled " The effects of exercise on FGF21 in adults: a systematic review and meta-analysis" aimed to summarise the effect of different exercises on FGF21 levels in adults to clarify the current knowledge in this area. I think this topic is important and worthy of attention. In total, the authors included 12 randomised controlled trials with 389 participants in their meta-analysis. The study may have good potential, but there are some inaccuracies, some missing information about the methodology, and the meta-analyses have many flaws. In this state, I do not recommend its publication.

The problems with the paper are as follows:

Formal shortcomings

On the face of it, the citations are wrong. I am not sure if the authors followed the author's instructions correctly. For example, the space between the text and the brackets. The number of authors in brackets is inconsistent. The overall impression of the citations is bad.

The text is not written according to the rules of academic writing. For example, this sentence: "Some shown maybe 67 the proteolytic cleavage to broke the form of FGF21 itself" does not seem to be scientific language.

Experimental design

The authors state that the meta-analysis followed the Preferred Reporting Items for Systematic Reviews and Meta-Analyses (PRISMA) statement and the PICO requirements. However, I dare to doubt this. The search strategy is not sufficient, especially the search formula is very weak to include all possible publications. For example, just by using the asterisk ("activit* (All Fields)") instead of the singular form with the suffix "Y" in the noun "activity", the WoS returns 80 thousand more possible papers. If authors really follow the PICO, where are the participants (adults with metabolic disorders) in the search terms? Overall, I recommend improving the search strategy.

There are more appropriate tools for quality assessment than Review Manager 5.4. I recommend to use RoB 2 for RCT in accordance with Cochrane.

The main concern is that the statistical tool was really wrongly chosen. It seems that all groups were merged. However, the Cochran estimator used by Revman does not allow robust variance estimation, which would be more appropriate in this case. The problem is the repeated use of control groups.

Overall, the use of this statistical tool is not appropriate even in the context of very different age groups and different types of interventions in terms of type of exercise, duration and intensity.

Validity of the findings

Because inappropriate analytical methods were chosen, I believe that the results estimated in this way are not correct at all.

Figure 1 is not the latest version of PRISMA flow diagram.

Leaving aside the generally inappropriate choice of analytical tools. The authors state that participants were blinded in about 30% of the trials (Figure 2)? This is ridiculous, how is it possible to blind people over weeks of exercise interventions? The trials included were not placebo-controlled trials.

Reviewer 3 ·

Basic reporting

The authors present a systematic review and meta-analysis regarding the effect of different modes of exercise on FGF21 levels in adults. Even though the paper contributes to the literature, I have several major issues. The language needs some editing; there are many errors regarding both spelling and grammar.

Experimental design

The acronyms must be specified the first time they appear in the text.
Abstract – what do you mean by the sentence “metabolic disorder adults with healthier state”?
Introduction –The introduction section needs to be improved. There is a lot of information, and their reading is challenging. Additionally, “multiple exercises” is not the best term for different types/modes of exercise.
Methods – Why was the strategy search limit from January 2005?
According to the eligibility criteria, the authors only included full-text studies and explicitly excluded published proceedings papers, non-full texts, and dissertations. This contradicts current recommendations for conducting systematic reviews and meta-analyses (see Cochrane recommendations) as it contributes to overestimating the results of published studies (publication bias). Furthermore, Cochrane recommends an additional search in the grey literature and in the reference lists of the included studies to identify potential eligible papers.
It is unclear which scale was used to assess the quality of studies and the risk of bias.
It is important to specify which measures (mean and standard deviation for example) were retrieved from the studies to run the meta-analysis.

Validity of the findings

Figure 1 shows that 12 studies were included in the qualitative synthesis. How many studies were included in the quantitative synthesis?
Please, correct the sentence in lines 150-2. This sentence gives the idea that all 389 participants have obesity, T2D, NAFLD, NASH, etc.
The information presented in lines 175-9 is confusing.
All the figures share the same subtitle.
Since the study aims to analyze the effects of physical exercise on FGF21 levels, the means and standard deviation of each group at baseline and follow-up (or the average difference between moments, direction of the effect, etc) would have to be analyzed. In the figures, we saw that only the mean and standard deviation of each group at a single point in time were reported, which only promotes comparison between groups, without considering the effects of the intervention. Please explain how the analyses were carried out, and whether or not they make it possible to analyze the effect of physical exercise on FGF21 levels.
Limitations – The sentence “we excluded studies with negative results” needs further clarification.

---

## Round 0.2 · accepted · Accept

The authors have addressed all of the reviewers' comments. The manuscript can acceptable for the publication.

Reviewer 1 ·

Basic reporting

I have no further comments.

Experimental design

I have no further comments.

Validity of the findings

I have no further comments.

Additional comments

I have no further comments.

Reviewer 2 ·

Basic reporting

no comment

Experimental design

no comment

Validity of the findings

no comment

Additional comments

I think the paper is now ready for publication.